# Genome-Wide Identification and Abiotic-Stress-Responsive Expression of *CKX* Gene Family in *Liriodendron chinense*

**DOI:** 10.3390/plants12112157

**Published:** 2023-05-30

**Authors:** Xiao Sun, Liming Zhu, Zhaodong Hao, Weihuang Wu, Lin Xu, Yun Yang, Jiaji Zhang, Ye Lu, Jisen Shi, Jinhui Chen

**Affiliations:** 1State Key Laboratory of Tree Genetics and Breeding, Co-Innovation Center for Sustainable Forestry in Southern China, Nanjing Forestry University, Nanjing 210037, China; sunxiao0075@163.com (X.S.); zhulm20160918@163.com (L.Z.); haozd@njfu.edu.cn (Z.H.);; 2Key Laboratory of Forest Genetics and Biotechnology, Ministry of Education, Nanjing Forestry University, Nanjing 210037, China

**Keywords:** *CKX* gene family, *Liriodendron chinense*, abiotic stress

## Abstract

*Liriodendron chinense* is a tree species of the *Magnoliaceae* family, an ancient relict plant mainly used for landscaping and timber production due to its excellent material properties and ornamental value. The cytokinin oxidase/dehydrogenase (CKX) enzyme regulates cytokinin levels and plays an important role in plant growth, development, and resistance. However, too-high or too-low temperatures or soil drought can limit the growth of *L. chinense*, representing a key issue for research. Here, we identified the *CKX* gene family in the *L. chinense* genome and examined its transcriptional responses to cold, drought, and heat stresses. A total of five *LcCKX* genes, distributed on four chromosomes and divided into three phylogenetic groups, were identified across the whole *L. chinense* genome. Further analysis showed that multiple hormone- and stress-responsive cis-acting elements are located in the promoter regions of *LcCKX*s, indicating a potential role of these *LcCKX*s in plant growth, development, and response to environmental stresses. Based on existing transcriptome data, *LcCKX*s, especially *LcCKX5*, were found to transcriptionally respond to cold, heat, and drought stresses. Furthermore, quantitative reverse-transcription PCR (qRT-PCR) showed that *LcCKX5* responds to drought stress in an ABA-dependent manner in stems and leaves and in an ABA-independent manner in roots. These results lay a foundation for functional research on *LcCKX* genes in the resistance breeding of the rare and endangered tree species of *L. chinense*.

## 1. Introduction

Cytokinins (CKs) are a group of plant hormones that play an important role in all aspects of plant growth and development, including apical dominance, stem or root branching, leaf spreading, lateral shoot growth, photosynthesis, seed germination, flower transition, and leaf senescence [1,2]. Cytokinins play opposite roles in shoot apical meristems (SAMs) and root apical meristems (RAMs). Plants with lowered cytokinin signal output or cytokinin content form larger RAMs and exhibit more rapidly growing roots [3,4]. Plant CKs promote developmental and physiological processes, drought tolerance, root architecture, and ultimately, crop productivity. CKs are degraded by cytokinin oxidases/dehydrogenases (CKXs), and CKs are precisely maintained in dynamic CK balance in nutritional tissues with the developmental regulation of the CKX-led irreversible degradation of catabolic CKs [5]. Studies have shown that the enhanced expression of *CKX* in roots to enhance cytokinin degradation leads to the formation of larger root systems in Arabidopsis, barley, oilseed rape, and rice [6].

The *CKX* gene family has been studied in a variety of plants, such as *Arabidopsis*, rice, wheat, tobacco, cotton, maize, soybean, chickpea, and others. For example, 7 *CKX*s (*AtCKX1*-*AtCKX7*) have been identified in *Arabidopsis*; a total of 13 *CKX*s (*ZmCKX1*-*ZmCKX13*) have been identified in maize; and 18 *CKX*s (*GmCKX01*-*GmCKX18*) have been identified in soybean [7,8]. Plants are affected by a variety of environmental abiotic factors, including internal flooding, soil salinity, temperature, and drought, which can interfere with all the metabolic activities of plants. The expression profiles of most miRNAs involved in plant growth and development are altered under abiotic and biotic stresses. These latter observations suggest that plant growth and development impaired under stress may be controlled by different stress-responsive miRNAs [9,10]. Recently, several reports have suggested that *CKX*s play an important role in stress response, especially under abiotic stress [11]. The expression profiles of *GmCKX* genes with respect to different stress treatments and tissues were different. Stress treatments significantly up-regulated the expression levels of *GmCKX13* in leaves and roots but down-regulated the expression levels of *GmCKX3* and *GmCKX8*; the *GmCKX14* gene was highly up-regulated in soybean leaves under three stress treatments, while its expression levels in roots were significantly down-regulated [11]. In a related study on tobacco, gene expression analysis revealed that each *NtCKX* gene responded differently to salt stress and exogenous abscisic acid treatment. The four *NtCKX* genes studied showed an ABA-induced expression trend with different peak times. Under salt stress, *NtCKX* expression was significantly repressed in two genes and up-regulated in the others [12].

Priyanka Jain et al. studied the expression pattern of *CKX* genes in wheat. *CKX7* was expressed at low levels in leaf, inflorescence, and spike tissues. *CKX4* and *CKX5* were expressed at higher levels in leaf tissues [13]. *CKX3* and *CKX11* were expressed at different levels in all the tissues studied, while *CKX10* was expressed at lower levels in all tissues except leaves [13]. *BjuCKX11* and *BjuCKX13* were significantly up-regulated in the S2 phase, which had the highest expression levels among the four periods, while the highest expression levels of *BjuCKX12* and *BjuCKX14* occurred in the S4 phase [14]. The expression of both genes in subpopulation V showed the same trend, i.e., it increased, then decreased, and then increased in the four periods (S1–S4), with a peak in S2 [14].

*Liriodendron chinense* (Hemsl.) Sarg. is a relict tree species that is native to southern China. It is a rare Tertiary relict tree species belonging to the magnolia family (Magnoliaceae) [15]. It is famous for its unique leaf shape, rapid growth, and soft texture, so it is often used as an ornamental tree and in wood production. However, the underlying genetic mechanisms of leaf development and morphogenesis remain poorly understood. As it is a valuable ornamental plant, it is meaningful to understand the development and morphogenesis of *L. chinense* leaves, and *L. chinense* genome and transcriptome information has recently been released [16]. Yang et al. performed transcriptome sequencing and comparative transcriptome analysis of the petals and sepals of *L. chinense*. in 2014 and found that carotenoid-biosynthesis-related genes were significantly differentially expressed between the petals and sepals of *L. chinense* [17]. Zhong et al. conducted a preliminary investigation of the genome of *L. chinense* rowanii in 2017 based on the Illumina sequencing platform, which showed that the genome of *L. chinense* rowanii belongs to a highly repetitive and highly heterozygous genome [18].

In this study, the genomic data of *L. chinense* were analyzed to identify members of the CKX gene family at the genome-wide level and to characterize their physical and chemical properties, basic features, gene structure, evolutionary relationships, chromosomal positioning, protein tertiary structure, and cis-acting elements. To determine the gene expression patterns of the *CKX* gene family members, we analyzed transcriptome data relative to different organs; somatic embryogenesis; and cold, heat, and drought stress conditions. Finally, quantitative reverse-transcription polymerase chain reaction (qRT-PCR) analysis further confirmed the differential expression patterns of *LcCKX5* genes in multiple organs and under drought abiotic stress. Evolutionary analysis helped to provide a comprehensive understanding of the origin and evolution of *CKX* in *L. chinensis* and laid the foundation for the study of its gene function. Many gene structures and promoter analyses also provided the basis for exploring their gene regulatory networks. This research targeted the study and transformation of abiotic-stress-related genes in the *L. chinense* genome, and the results provide a valuable basis for further functional studies of *LcCKX* in the context of cytokinin dynamic balance and abiotic stress, which could help improve adaptation to drought environments and lay the foundation for the further development of the quality-timber industry.

## 2. Results

### 2.1. Identification of CKX Gene Family in L. chinense Genome

A total of five *CKX* genes were identified in the genome of *L. chinense* and were named *LcCKX1*-*LcCKX5* according to the nomenclature of the *CKX* gene family in *Arabidopsis* (Table 1 and Appendix A). CDD analysis showed that these five LcCKX proteins all possess two domains, i.e., an FAD-binding domain and a cytokin-binding domain. Physicochemical property analysis showed that the molecular weight of these five CKX proteins ranged from 45,216.77 to 59,699.25, while the isoelectric point ranged from 5.7 to 8.8. In addition, the hydrophilic index (GRAVY) of LcCKXs was less than 0, except for LcCKX2, indicating that most LcCKXs are mostly hydrophilic proteins.

### 2.2. Phylogenetic Analysis of LcCKX Proteins

To further explore the phylogenetic relationships and evolutionary history of the *LcCKX* gene family, we constructed a phylogenetic tree containing CKX proteins from the basal angiosperm *Amborella trichopoda*, the magnoliophyte *L. chinense*, the monocotyledons *Oryza sativa* and *Zea mays*, and the dicotyledon *Arabidopsis thaliana*. A total of 44 CKXs were used to construct a phylogenetic evolutionary tree using the maximum-likelihood (ML) method, leading to three main phylogenetic groups (I, II, and III), two of which could be further divided into two subgroups (IIa and IIb, and IIIa and IIIb) (Figure 1). Among them, group I comprises only three AmCKXs, while the rest of the groups contain CKXs from all five examined species. Specifically, LcCKX1 and -2 were grouped in IIIb, and LcCKX3, -4, and -5 were grouped in II (Figure 1).

### 2.3. Analysis of Structure and Conserved Motifs of LcCKX Genes

Based on the analysis of the gene structure (Figure 2), we found that LcCKX1, LcCKX2, and LcCKX5 have four exons, while LcCKX3 and LcCKX4 have six and five exons, respectively. The identification of conserved motifs showed that all five LcCKXs protein sequences contain motifs 1, 3, 4, 6, 8, and 9 in the same order, hinting at the conserved function of these LcCKX proteins. Meanwhile, motifs 2 and 7 were only absent in LcCKX2 and -7, respectively, while motif 5 was present in LcCKX3-5, implying the functional differentiation of these LcCKXs.

### 2.4. Chromosome Localization and Tertiary Structure of LcCKX Genes

The chromosomal localization analysis showed that members of the *LcCKX* gene family are unevenly distributed on 13 *L. chinense* chromosomes (Figure 2a). Four *CKX* genes are distributed on three chromosomes, i.e., Chr2, -6, and -7, and the remaining one is located on a scaffold. The tertiary structure of a protein is the three-dimensional conformation of a protein molecule in its natural folded state. The tertiary structure is formed by further coiling and folding on the basis of the secondary structure, and the characterization of the tertiary structure of proteins is of great importance for the study of the functional properties of proteins. The predictions of the 3D structures of LcCKXs (Figure 3b) show that LcCKX1 and LcCKX2 have a heart-shaped structure and a smaller mass. The LcCKX3, LcCKX4, and LcCKX5 models are more similar in structure and have a butterfly shape, and the protein structure morphologies show a high degree of similarity. These proteins all have one or several grooved structures. The grooved structures may provide the conditions for them to bind substrates to exert enzymatic activity, and the center of enzymatic activity may be located in these structures.

### 2.5. Prediction of Cis-Acting Elements in LcCKX Promoter Regions

The cis-acting elements in the gene promoter region can be bound by specific transcription factors, thus regulating the expression of downstream genes. The prediction of cis-acting elements suggested that there are three main classes in the *LcCKX* family (Table 2), i.e., phytohormone signaling, environmental stress, and MYB binding sites. All *LcCKX* promoters contain abscisic acid (ABA) response elements, indicating that the *LcCKX* gene family might be sensitive to ABA. However, part of the *LcCKX* gene family also potentially responds to other phytohormones, such as ZT, MeJA, GA, and SA, indicating that these *LcCKX*s might be extensively involved in plant growth and development. Furthermore, most *LcCKX*s contain cis-elements that are related to cold and/or drought stress, implying a potential function of these *LcCKX*s in abiotic stress response.

### 2.6. Gene Expression Pattern Analysis of LcCKX Family

The transcriptome sequence data of *L. chinense* were downloaded from the NCBI SRA database, and the expression level of the genes was obtained using salmonid analysis. We determined the expression patterns of the *LcCKX* gene family in leaves under drought, cold (4 °C), and heat (40 °C) stresses (Figure 4a). *LcCKX5* showed a strong transcriptional response under all three stresses. Specifically, the expression of *LcCKX5* was up-regulated and peaked after 12 h and 3 d in response to heat and drought stresses, respectively. In contrast, the expression level of *LcCKX5* decreased under cold stress. In comparison, *LcCKX1*, -*3*, and -*4* responded to these abiotic stresses, but less strongly, while *LcCKX2* seemed not to be expressed at all. The above results indicate that *LcCKX* genes differentially responded to different stresses and that *LcCKX5* might be a valuable target for further research on the resistance breeding of *L. chinense.*

To elucidate the expression pattern of *LcCKX*s in the growth and development of *L. chinense*, we constructed gene expression profiles for different stages of the somatic embryogenesis of *L. chinense* (Figure 4b). *LcCKX2*, *LcCKX3*, and *LcCKX5* were found to be involved in the process from embryonic callus to regeneration plantlet; *LcCKX3* showed a transcriptional association during somatic embryo development, while *LcCKX5* might be involved in somatic embryo maturation.

Meanwhile, when comparing the expression of the *LcCKX* gene family in different organs of *L. chinense* (Figure 4c), the analysis showed that *LcCKX1* was highly expressed in different organs of the petals, especially in the shoot apex, while *LcCKX2* was more expressed in bracts and stamens than in other parts. In contrast, *LcCKX3* and *LcCKX4* had low or no expression in different tissues. The above results suggest that *LcCKX* genes are extensively involved in plant growth and development, as well as responses to abiotic stresses, in *L. chinense*.

### 2.7. Expression Analysis of LcCKX5 under Abiotic Stress

To further investigate the expression pattern of *LcCKX5* under drought stress, 20% PEG8000 was used to treat *L. chinensis* seedlings to simulate drought conditions, and the transcriptional response of *LcCKX5* was quantified using qRT-PCR. The internal reference genes and related primers are shown in the Table 3. The results show that the expression change of *LcCKX5* differed between roots and shoots in response to drought stress. Specifically, under drought stress, the expression of *LcCKX5* decreased and then increased over time in roots, while there were no significant changes in pairs of stems, and there was a gradual decrease in leaves (Figure 5 PEG). Increasing ABA treatment in parallel with drought stress showed a decreasing trend in *LcCKX5* in roots for a short period of time which subsequently appeared to increase (Figure 5 PEG+ABA-Root); while overall expression was suppressed in stems and leaves, no significant effect on roots under drought stress was observed (Figure 5 PEG+ABA-Stem and PEG+ABA-Leaf). Interestingly, the exogenous application of the ABA biosynthesis inhibitor Fluidon (Flu) completely disrupted the expression pattern of *LcCKX5* in both stems and leaves while still showing a decreasing trend and then an increasing trend in roots under drought stress (Figure 5 PEG+Flu). The expression of *LcCKX5* decreased and then increased with the increase in stress time in roots and stems, while the opposite was true in leaves.

## 3. Discussion

Cytokinin dehydrogenase regulates the content of cytokinin and plays an important role in balancing the synthesis and degradation of cytokinin in plants [19,20]. *CKX* plays a very important role in balancing the synthesis and degradation of cytokinins in plants. With the continuous development of plant genomics and reverse genetics, we can provide powerful tools to study *CKX* genes [21]. The decipherment of the genome of *L. chinensis*, a representative species of the Liriodendron genus in the Magnoliophyta family, provides new insights into the phylogenetic position of magnoliophytes in angiosperms [22]. *CKX* gene families have been identified or deduced in *Arabidopsis*, rice, wheat (*Triticum aestivum* L.), maize (*Zea mays* L.), cotton (*Gossypium hirsutum* L.), and alfalfa (*Medicago sativa* L.) [23,24,25]. The *CKX* gene family was identified or deduced in different plants, such as Arabidopsis, maize, cotton, and alfalfa. There are 7 *AtCKX* genes in *Arabidopsis*, 13 *ZmCKX*s in maize, 11 *OsCKX*s in rice, 8 *VvCKX*s in grape (Vitis vinifera L.), and 5 *MnCKX*s in mulberry (Morus notabilis) [7,26,27,28,29]. In this study, five *LcCKX* genes were identified in L. chinensis. The number of genes in other species was small, which showed that *CKX* is a gene family with a small number of members. The five *LcCKX* genes identified fall into three subgroups, and the genes in each subgroup are highly conserved in terms of protein physicochemical properties, conserved motifs, and gene structure.

The identification of members of a species-specific gene family using the conserved structural domains of a particular gene family is an accurate and efficient method. Analysis of the conserved structural domains showed that most of them are relatively conserved in the family genes, and only some of them show some specificity; for example, motif 5 is only present in group II (*LcCKX3~5*) and motif 10 is only present in group IIa (*LcCKX3~4*). This specificity may be related to the functions assumed by different groups of *CKX* and to the differences in the higher structure of the proteins. Analysis of the 3D structure of the proteins showed that the butterfly-shaped protein structure corresponds to three members of group II, while the protein with a heart-shaped tertiary structure corresponds to two members of group I.

Cis-regulatory elements are specific motifs located in the promoter region of genes that act as binding sites for genes and play an important role in stress response by regulating the transcription of downstream genes [30]. In addition, some phytohormones, such as abscisic acid, salicylic acid, jasmonic acid, and ethylene, are also involved in regulating the adaptive response of plants to abiotic stresses [31]. According to the results of the predicted cis-acting elements, the *LcCKX* gene mainly responds to plant hormones, light, and abiotic stress. In other plant species, such as in maize, these cis-acting elements are also present in large numbers in the promoter regions of the CKX gene [32]. These results further suggest that the *LcCKX* gene may be important and involved in the plant response to abiotic stress. As the promoter regions of the five *LcCKX*s contain different types, numbers, and distributions of cis-acting elements, but also share the same characteristics, it is assumed that the different *LcCKX*s have different expression patterns.

The cytokinin group of plant hormones is involved in regulating several aspects of plant growth and development [33], many of which have direct effects on crop improvement, such as the regulation of the root crown structure [34], the regulation of inflorescence meristem tissue activity and seed yield [35,36], the regulation of leaf senescence and photosynthesis, and response to biotic and abiotic stresses [27,37,38]. According to the RNA-Seq data, under abiotic stress conditions such as drought, low-temperature treatments, and high-temperature treatments, the expression of some *LcCKX* genes can be changed, thus playing a defensive and protective role in plant growth and development processes, which may also be related to the resistance of goosefoot, but the exact mechanism of this effect needs to be studied in the future.

In other species, *CKX*s have diverse expression patterns, even when they are tandemly repeated. In maize, for example, *ZmCKX1*, *ZmCKX6*, and *ZmCKX10* are expressed in all tissues, while *ZmCKX7* and *ZmCKX8* are only expressed in male ears and are clearly tissue-specific. *ZmCKX*s differ in number and expression in different tissues under the same stress conditions [32]. In *Plasmodiophora brassicae*, *CKX*s show different expression patterns in different tissues, indicating different functions [39]. Under normal conditions, three of these genes were highly expressed in all tissues studied, which is characteristic of the expression pattern of housekeeping genes [39].

Among tree species, *CKX*s have different expression patterns in different tissues. In studies related to *CKX* in *Jatropha curcas*, it was found that *JcCKX1* is mainly expressed in flower buds, roots, and female flowers, while *JcCKX2* shows very strong expression in female flowers and seeds. *JcCKX3* is highly expressed in male flowers, and *JcCKX4* shows high expression levels in mature leaves and female flowers, and extremely high expression in seeds. *JcCKX5* is mainly expressed in stems, tender leaves, and fruits, while *JcCKX6* is expressed in all tissues, and *JcCKX7* is mainly expressed in roots [40]. In *Malus domestica*, the expression of *CKX* is significantly higher in leaves than in other tissues; *MdCKX1*, *MdCKX7*, *MdCKX9*, and *MdCKX11*/*12* have high levels of expression in roots. In addition, compared with other tissues, all *MdCKX* levels are lower in the stem and axillary buds, indicating that CK accumulation is naturally limited to the stem and axillary buds. At the same time, Ming Tan et al. treated the experimental materials with 6-BA, and throughout the sampling time, *MdCKX1*, *MdCKX2*, *MdCKX5*, and *MdCKX10* were significantly up-regulated, especially *MdCKX10*, whose transcription level increased by more than 100 times in 48 h. The *MdCKX2*, *MdCKX4*, *MdCKX6*, *MdCKX7*, and *MdCKX8* transcripts did not respond to 6-BA treatment, and their expression levels were similar to those in untreated control buds after 24 and 48 h [41]. Combined with the relevant results of this study, it can be found that the expression pattern of *CKX* in different tissues of trees is similar and that there are also different expression locations. In general, the expression amount of *CKX* in roots is relative, while under non-biotic stress, the response of *CKX* in roots is faster and the response in leaves and stems is slower.

In summary, this research focused on *LcCKX5* RNA-Seq data, namely the expression pattern of *LcCKX*s with respect to different tissues, time, and stress conditions, as well as functional predictions. The results could help to screen potential resistance genes and provide very detailed and reliable information for subsequent studies.

## 4. Materials and Methods

### 4.1. Identification of CKX Genes in L. chinense

*L. chinense* protein sequences were downloaded from NCBI (https://www.ncbi.nlm.nih.gov/, accessed on 12 April 2022). The Hidden Markov Model (HMM) profile (PF09265) of the cytokinin-binding conserved structural domain of the *CKX* gene family and the HMM profile (PF01565) of the FAD-binding conserved structural domain were downloaded from the Pfam database (https://pfam.xfam.org, accessed on 5 May 2022). HMMER 3.2 software was used to analyze and identify the protein sequences that contain both conserved structural domains. The candidate member sequences were compared using NCBI (https://www.ncbi.nlm.nih.gov/, accessed on 5 July 2022) to determine their CDS and position information, and the CDD database (https://www.ncbi.nlm.nih.gov/cdd/, accessed on 5 July 2022) was used to manually analyze and confirm that the candidate genes contained 2 conserved structural domains at the same time; finally, the members of the *CKX* gene family were obtained.

### 4.2. Protein Physicochemical Properties and Tertiary Structure Prediction of CKX Genes in L. chinense

According to the identified *CKX* gene ID, the chromosome location information was obtained from the GFF3 file. The physicochemical properties of the proteins, including relative molecular mass, theoretical isoelectric point, instability index, and hydrophilicity, were analyzed using the web-based online tool ExPASy (https://web.expasy.org/, accessed on 5 July 2022). The position information of the *LcCKX* genes on the chromosome was picked up from the *L. chinense* annotations using TBtools. The predictive analysis of the 3D structures of different members of the *CKX* gene family was performed according to the online website SWISS-MODEL (https://swissmodel.expasy.org/, accessed on 5 July 2022).

### 4.3. Phylogenetic Analysis of CKX Genes in L. chinense

MEGA v10.1.8 (Temple, Philadelphia, PA, USA) was used to examine *CXK*s from *L. chinense*, rice, and *Arabidopsis* to determine their phylogenetic relationship. We used MUSCLE implemented in MEGA to align the amino acid sequences and the maximum-likelihood estimation algorithm to create phylogenetic trees with a bootstrap value of 1000. DANMAN v9.0 (Lynnon Corporation, San Ramon, CA, USA) was used for the multi-fragment alignment of amino acid sequences.

### 4.4. Gene Structure and Conserved Motif Analysis of CKX Genes in L. chinense

The structure of the gene was produced based on the length of the *CKX* gene and the location information of the CDS. The conserved patterns of the aa sequences of all members of *LcCKX* were analyzed using MEME (http://memesuite.org/tools/meme/, accessed on 5 July 2022), and the phylogenetic tree, gene structure, and conserved patterns of the *LcCKX* gene were merged using TBtools.

### 4.5. Cis-Acting Elements of CKX Genes in L. chinensis

The genomic DNA sequences 2000 bp upstream of TAG in the sequences of different members of the *LcCKX* family were selected using whole-genome data of *L. chinense*, predicted and analyzed in cis using Plant Care (http://bioinformatics.psb.ugent.be/webtools/plantcare/html/, accessed on 5 July 2022), and mapped and displayed using OriginPro9.06Bit software to predict gene function.

### 4.6. RNA-seq Analysis of LcCKX Gene Expression Levels in Different Organs and under Multiple Stresses

The drought stress transcriptome data of L. hybrid were annotated with accession number PRJNA679101 and can be downloaded from NCBI (https://www.ncbi.nlm.nih.gov/bioproject/PRJNA679101/; accessed on 5 September 2022). The data on somatic embryogenesis are unpublished. Heatmaps visualizing expressions were realized with TBtools software (v1.09). The log2 (TPM + 1) value was used for standardization and hierarchical cluster analysis.

Transcript data of different organs and hybrid data of *L. chinense* under high-temperature stress and drought stress were downloaded from NCBI. The transcript data on low temperature, *L. chinense* petal development, and hybrid *L. chinense* somatic embryogenesis are undisclosed data. The expression levels of related genes are listed in Appendix A. All mRNA abundance values were measured in transcripts per million (TPM) based on the *L. chinense* genomic database.

### 4.7. Plant Materials and Abiotic Stress Treatment

Seedlings produced during the somatic embryogenesis of *L. chinense* were used as the starting material for this study. Prior to stress treatment, somatic embryogenic seedlings were removed from the culture vessels and domesticated in a greenhouse for 2 weeks (conditions: 22 °C, prolonged light for 16 h and darkness for 8 h, and relative humidity of 75%). Three biological replications were performed using 15% PEG8000 to simulate a natural drought environment. Roots, stems, and leaves were harvested after 6 h and 24 h of drought treatment, respectively. All experimental tissue samples were immediately frozen in liquid nitrogen and then stored at −80 °C.

### 4.8. RNA Extraction and Quantitative Real-Time PCR Analysis

The first-strand gene was synthesized from 1.0 mg of RNA using an Evo M-MLV RT kit with gDNA Clean (Changsha Precision Biotechnology (Hunan) Co.). An Equalbit 1× dsDNA HS Assay Kit (EQ121-01; Vazyme, Nanjing, China) was used to complete the quantification of all reverse cDNA. Polymerase chain reaction amplification was performed in 20 µL using a SYBR^®^Green PreMix Pro Taq HS qPCR Kit (Precision Biotechnology (Hunan) Co., Ltd., Changsha, China) using a Roche LightCycler^®^ 480 Real-Time Polymerase Chain Reaction System. Three replicates were performed for each selected gene. The expression pattern of CKX-related genes under low-temperature stress was studied using qRT-PCR, and Primer 5.0 software was used to design primers in the non-conserved structural domain region of the genes. qPCR SYBR Green Master Mix (Vazyme) was used for real-time quantitative PCR, and *GAPDH* and *18s* were used as the internal control genes. The real-time PCR cycling parameters were 95 °C for 30 s, followed by 45 cycles at 95 °C for 5 s and 60 °C for 30 s, with melting curve analysis. All reactions were performed in triplicate to ensure the repeatability of the results. Gene expression levels were calculated using 2^−∆∆Ct^ [42].

## 5. Conclusions

In this study, comprehensive analysis of the *CKX* gene family in the *L. chinense* genome was conducted, and five *LcCKX* genes were identified. Subsequently, gene structure analysis, phylogeny, chromosomal localization, gene duplication, and genome-wide identification and analysis of *CKX* family genes in *L. chinense* were carried out using bioinformatics and qRT-PCR. The differential expression of *LcCKX5* genes in different tissues of *L. chinense* and the different expression trends under drought stress and ABA treatment may indicate that they play an important role in drought resistance and tissue development. The present study provides comprehensive information on the *CKX* genes in *L. chinense*, which could help to determine the functions of *CKX* genes.

## Figures and Tables

**Figure 1 plants-12-02157-f001:**
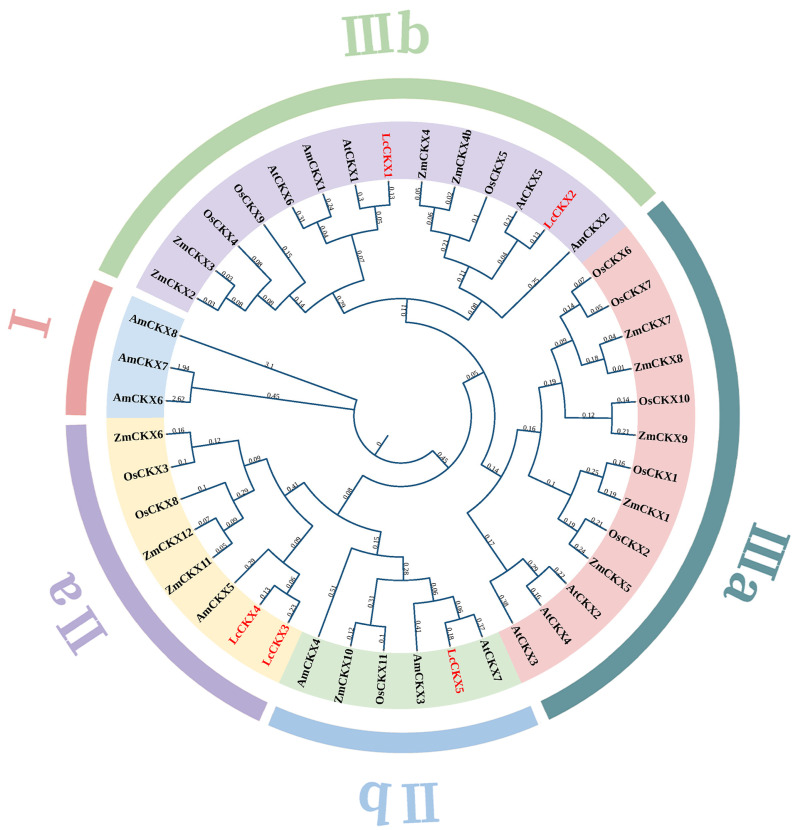
Phylogenetic analysis of CKX proteins. The amino acid sequences of CKXs were aligned using Clustal X, and the phylogenetic tree was constructed using the maximum-likelihood (ML) method in MEGA11 with 1000 bootstrap replicates. The cytokinin dehydrogenase/oxidase genes in *L. chinense* are marked in red. At: *Arabidopsis thaliana*; Am: *Amborella trichopoda*; Lc: *Liriodendron chinense*; Os; *Oryza sativa*; Zm: *Zea mays*.

**Figure 2 plants-12-02157-f002:**
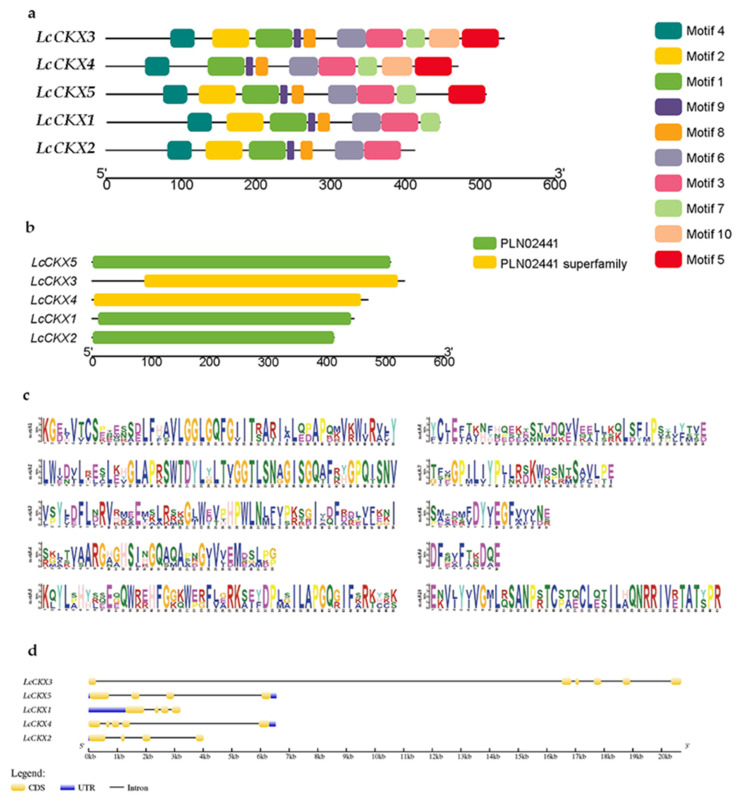
Gene structure and conserved motifs in LcCKXs. Phylogenetic relationships (**a**), conserved motifs (**b**), amino acid composition of each motif (**c**), and gene structure (**d**) of *CKX* genes of *L. chinense*. Differently colored boxes represent different themes and their positions in each *LcCKX* sequence.

**Figure 3 plants-12-02157-f003:**
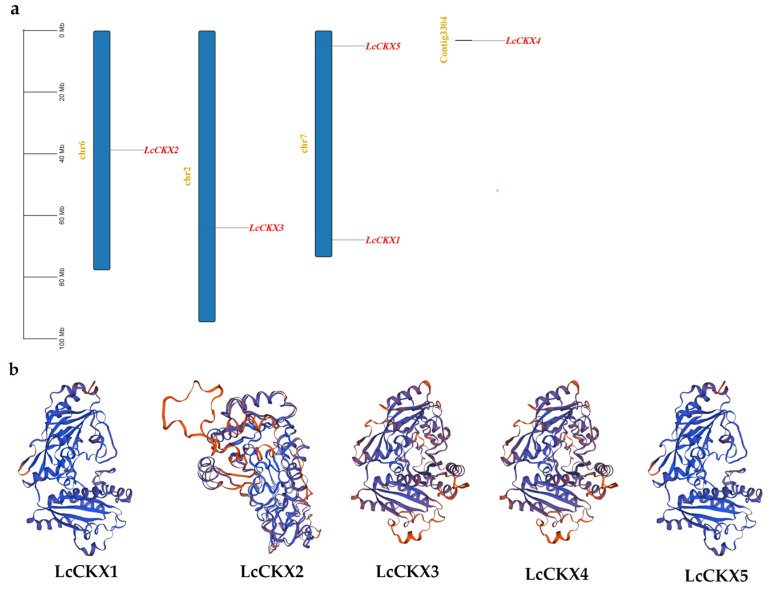
Chromosomal distribution of the *CKX* genes of *L. chinense* (**a**) and 3D structures of LcCKX oxidases showing functional sites (**b**). a: The scale located on the left panel is in bases, indicating chromosome sizes. The chromosome number is indicated on the left of each chromosome.

**Figure 4 plants-12-02157-f004:**
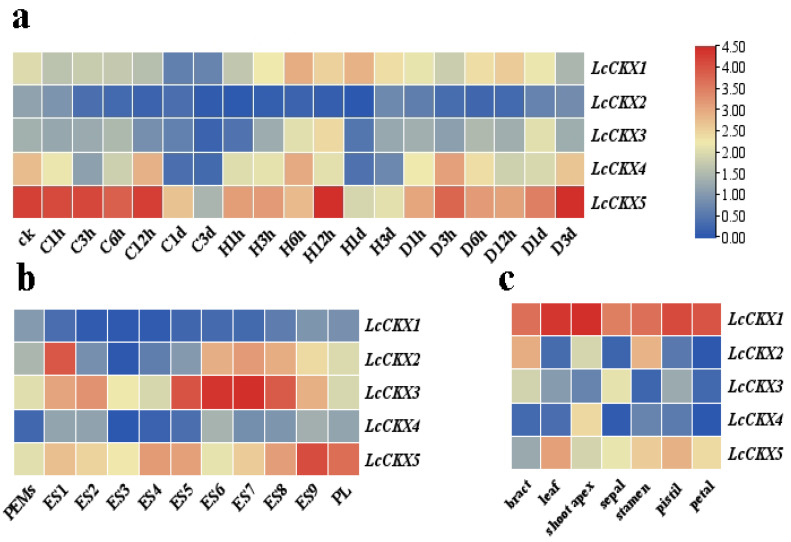
*LcCKX* gene expression profiles in different organs. (**a**): Heatmap of *LcCKX* genes under drought, cold and heat stresses. (**b**): Heatmap of *LcCKX* genes in somatic embryogenesis (**c**): Heatmap of *LcCKX* genes in seven tissues. The heatmaps show the means of three biological replicates. Transcripts per million (TPM) was used to indicate the gene expression level. PEM: embryogenic callus; ES1: 10 days after liquid culture; ES2: 2 days after screening; ES3: ABA treatment for 1 day; ES4: ABA treatment for 3 days; ES5: globular embryo; ES6: heart-shaped embryo; ES7: torpedo embryo; ES8: immature cotyledon embryo; ES9: mature cotyledon embryo; PL: plantlet.

**Figure 5 plants-12-02157-f005:**
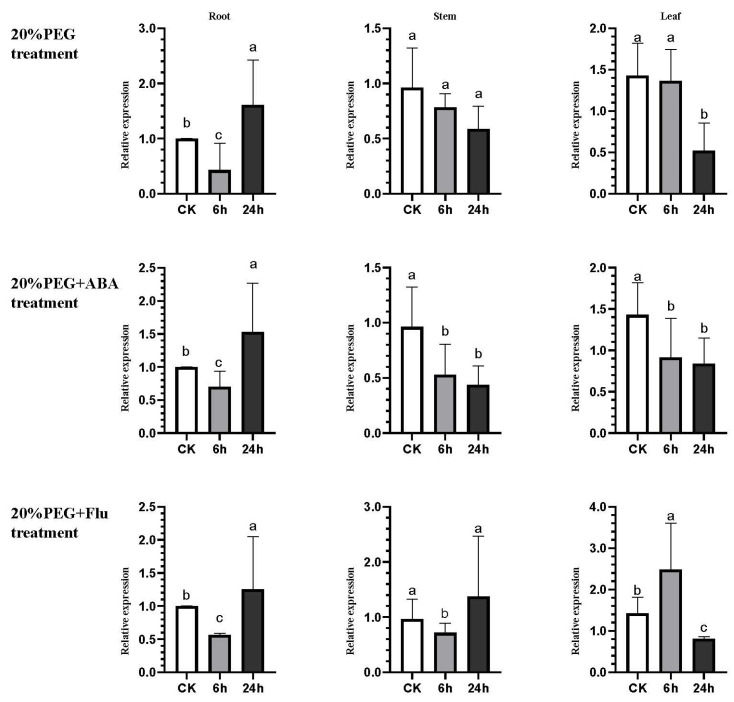
Expression profiles of *LcCKX5* under abiotic stress. Values with a different letter (**a**–**c**) were significantly different when assessed using Duncan’s multiple range test (*p* < 0.05). Same treatment per row, same tissue site per column. The 20% PEG treatment, 20% PEG+ABA treatment, and 20% PEG+Flu treatment in the figure were referred to as PEG, PEG+ABA, and PEG+Flu, respectively.

**Table 1 plants-12-02157-t001:** Summary of *L. chinense CKX* gene family members.

Gene Name	Gene ID	Length (aa)	Molecular Weight (kDa)	Isoelectric Point (pl)	GRAVY
*LcCKX1*	*Lchi24464*	446	49,620.79	8.8	−0.136
*LcCKX2*	*Lchi34027*	412	45,216.77	6.1	0.044
*LcCKX3*	*Lchi12502*	532	59,699.25	6.64	−0.196
*LcCKX4*	*Lchi32870*	470	53,131.66	5.98	−0.164
*LcCKX5*	*Lchi19652*	508	56,314.36	5.7	−0.089

**Table 2 plants-12-02157-t002:** *LcCKX* promoter cis-element analysis.

Gene Name	Plant Hormone					Environmental Stress				MYB Binding Site
	ABA	ZT	MeJA	GA	SA	Light	Defense	Circadian	Low Temperature	Drought Inducibility
*LcCKX1*	√	√	√	√		√			√	
*LcCKX2*	√				√	√		√		√
*LcCKX3*	√	√	√		√	√			√	√
*LcCKX4*	√		√	√		√	√		√	√
*LcCKX5*	√	√		√		√	√		√	

**Table 3 plants-12-02157-t003:** qRT-PCR primers used to quantify *LcCKX5* gene expression.

Gene Name		qRT-PCR Primer
*LcCKX5*	R	AAATGGCCTTCCTCTCGACG
	F	CTTCGTTTCGGCCGTTCATC
*18S*	R	CTGCCTTCCTTGGATGTGGT
	F	GCCCGTCGCTCTGATGAT
*Acting97*	R	TGGTCGCACAACTGGTATCG
	F	TTCCCGTTCAGCAGTGGTCG

## Data Availability

Transcriptome data of somatic embryogenesis and tissues have not yet been published. The drought stress transcriptome data of L. hybrid were annotated with accession number PRJNA679101 and can be downloaded from NCBI (https://www.ncbi.nlm.nih.gov/bioproject/PRJNA679101/; accessed on 5 September 2022). The complete genome, transcript/protein sequences, and genome feature file of Lchi were downloaded from https://www.ncbi.nlm.nih.gov/assembly/GCA_003013855.2 (accessed on 5 September 2022).

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
