# Peer review of "Genome-Wide Identification and Abiotic-Stress-Responsive Expression of CKX Gene Family in Liriodendron chinense"

_plants, 2023, doi:10.3390/plants12112157_

Round 1

Reviewer 1 Report

Manuscript “”Genome-wide ….. chinense” identified CKX gene family and studied its response under different abiotic stresses. The study is well executed and presented very professorially. Paper may accept in its present form.

Reviewer 2 Report

The authors must seek the help of a scientific editor to help them produce a viable manuscript for publication. In addition, there is a need to provide a more nuanced introduction and justification of the study, with clear objectives.  The materials and methods are sketchy and do not meet the normal standards of repeatability i.e., providing such detail and clarity that other researchers can repeat the study and validate the results of this study or otherwise.  The presentation of the results and their discussion are similarly obtuse and require major revision.

  The introduction resembles that of a review article and not that of a research article. What’s the gap of knowledge? Which is the scope of the manuscript? What hypothesis have been made? The introduction should be revised accordingly.

Experimental section:. A more succinic yet complete writing should be done. Moreover the author state that a statistical analysis has been made. I believe that the authors should give more details about the analysis performed.

The scientific background of the topic is poor. In "Introduction" and "Discussion", the authors should cite recent references between 2016-2022 from JCR journals.

A. Khan, D.K.Y.Tan, M.Z.Afridi, H.Luo, S.A.Tung , M. Ajab, Fahad S (2017) Nitrogen fertility and abiotic stresses management in cotton crop: a review, Environmental Science and Pollution Researchdoi:10.1007/s11356-017-8920-x

A. Noman, Fahad S, M. Aqeel, U. Ali, Amanullah, S. Anwar, S. Khan,M. Zainab. (2017). miRNAs: Major modulators for crop growth and development under abiotic stresses. Biotechnol Letter DOI 10.1007/s10529-017-2302-9

English should improve by a native person. The paper suffers from a poor English structure throughout and cannot be published or reviewed properly in the current format. The manuscript requires a thorough proofread by a native person whose first language is English. 

Reviewer 3 Report

It is not well explained why authors focused on the analysis of the CKX gene with respect to Liriodendron chinense as rare and endangered tree species.

Necessary for chromosome number and genome information in Introduction.

It is necessary for more discussion on expression pattern of CKXs in the comparison with other species relating to tree species.

Scientific names of plants should only be used once at the first place they appear.

Pay attention to the use of abbreviations.

Figure 5    no description of Figure 5-a, 5-b, 5-c

L201   ABA treatment -----   Is its content OK?

L203   Is its content OK as well?

L208   drought stress

L212  Cytokine oxidase/dehdrogenase

L132  chromosomal

Round 2

Reviewer 2 Report

Accepted as it stands